# Once More, with Feeling! Digital Campaigns and Emotional Candidacies in X in Andalusia and Castilla y León

**Sergio Pérez-Castaños** [1],* and **Juana María Ruiloba-Núñez** [2]

[1] Faculty of Law, Universidad de Burgos, 09002 Burgos, Spain
[2] Faculty of Law, Department of Public Law, Universidad Pablo de Olavide, 41013 Sevilla, Spain; jruiloba@upo.es
* Correspondence: spcastanos@ubu.es

**Abstract:** In 2022, two regional elections were held that initiated a new electoral cycle in Spain. The first, in February, took place in Castilla y León where, for the first time, the extreme right-wing party, VOX, formed a coalition in a regional government with the right-wing Partido Popular, while, simultaneously, the extreme left-wing party Podemos was barely left with parliamentary representation. The second, in June in Andalusia, led to VOX increasing its representation by two seats and beginning its national level electoral growth; the far-left that was split in two parties, finding that the dynamic that began in Castilla y León also affected them as they lost 10 representatives. This article seeks to compare the emotional strategies of these extremist parties in the 2022 regional elections, focusing especially on leaders' and parties' communication strategies on X, formerly Twitter, during the electoral campaign. We expect to find differences between political parties but also between regions. To test this, we carried out an 'X account content analysis'. We selected those accounts representing the candidates to the regional President and also the main political party accounts in the regions: namely VOX in both regions, Podemos in Castilla y León, and Adelante Andalucía and Por Andalucía in Andalusia.

**Keywords:** emotions; political communication; X; political polarization; Spain





## 1. Introduction

Emotions and their application in the context of electoral campaigns have been an interesting and widespread field of study from the perspective of Political Science, Political Communication and Political Psychology since the end of the last century (Marcus 2000). Since then, the growth in new technologies and their applicability to electoral campaigns has become a common focus of analysis—although the measurement of emotions in campaign scenarios has generated differences in the academic community (Lynggaard 2019).

Recently, the most prominent studies have analyzed the role of emotions in politics (Corduneanu 2018; Gravante and Poma 2018; Caramelo Pérez 2020; Villa-Gómez et al. 2020), focusing mainly on how emotional messages were transmitted to citizens (Pereira et al. 2021; Rivera-Otero et al. 2021b; Oñate et al. 2022) or, more specifically, on how the different social networks act as a sounding board for political ideas and, therefore, for the emotions that are transmitted and processed (Jaráiz Gulías et al. 2020, 2021; Rivera-Otero et al. 2021a; Törnberg 2022). One of the driving elements in this regard is the role of leaders as both mobilizers and channelers of emotions (Ferreira da Silva et al. 2021). This is because emotions have a special property that allows them to be transferred from one element to another: that is, an image, a tune or a person pervades the container of information (van Hoef and O'Connor 2019).

Undoubtedly, electoral campaigns are the battleground where emotions are most clearly reflected (Caramelo Pérez 2020; Grüning and Schubert 2022). Consequently, efforts are currently focused on understanding the cognitive processes involved in the exposure to certain emotions and the changes that these can cause in the way society perceives reality (Marcus 2000; Grüning and Schubert 2022).

Considering all these issues related to the importance of emotions in politics in general and focusing on the electoral campaign in particular, in the next section we will consider how to structure the analysis of emotional discourse of political parties that are ideologically located in more extreme positions, of which authors such as Vittori (2017) and Rooduijn et al. (2019) have classified as 'populist'. In Spain, these are the populist radical right party VOX[1], which has the same denomination in the two regions under comparison; at the other extreme, however, are Podemos and the different denominations and splits that occurred in Andalusia; that is, Adelante Andalucía (AA) and Por Andalucía (PORA). In Castilla y León, this organisation coincided in denomination with the Unidas Podemos (UP) electoral coalition.

It Is essential to study the role that the electoral campaign plays in the transmission of emotions (Ruiz-Sánchez and Alcántara-Plá 2019). All the more so in the context of generalized disinformation, much of which is fostered by social networks (Fernández and Rodríguez-Vigil 2019). In this sense, it seems appropriate to focus our analysis on X, formerly Twitter, as it is currently one of the most widely used tools and several studies in Spain have already shown that the transmission of emotional content on X is greater when talking about politics (Jaráiz Gulías et al. 2021; Agüero-Merino 2023). The political parties in our analysis have been chosen as they represent extreme political positions on the ideological spectrum. It should be noted that this typology of parties is currently in flux, as radical left parties are currently in a period of decline while VOX has been on an upward trend in recent elections.

Thus, the data collected show that there are subtle differences between organisations and between regions. In Castilla y León, there are differences in the emotional content, which is more present in Podemos, while in the case of Andalusia it is VOX that has a slightly higher emotional content. The same applies to the negative and humorous content of attacks on other parties, people or candidates. On the other hand, positive emotions have a greater weight in left-wing parties than in those of the right. These findings, which will be detailed in the following sections, are limited by the fact that it is not possible to know the impact of the communications and content, as X's change of ownership has meant that these services are now paid services.

## 2. Emotions and Politics: A Storied Journey

As mentioned, our analysis will focus on the organisations on the extreme left and right of the ideological spectrum, i.e., Podemos—and its split in Andalusia—and VOX. These two populist parties—despite the differences that might exist between them (Rooduijn et al. 2019; Vampa 2020; Caravantes 2020)—have a similar communicative dynamic, to the extent that populism usually seeks to favor an emotional discourse that links the postulates of the parties to any social profile within its reach (Vittori 2017; García-Hípola et al. 2021). In this sense, it is relevant to analyze how two organisations from different ends of the ideological spectrum, and with opposing electoral expectations, try to reach citizens. Even more so if we consider that, in the case of Andalusia, the left-wing party is divided into two groups that are unable to capture the electorate's vote.

In this context, the study of social networks acquires fundamental importance to understand the use of emotions in this new sphere. Our measurement focuses on monitoring X due to its wide use among certain age brackets and its outstanding activity as a generator and disseminator of political information (Calderón-Monge 2017; Larsson 2020). This paper presents an analysis of the content of the social networks of both political leaders, understood as the people who head the electoral lists of the different political organisations, and the main political parties in the contest. Since the greatest point of emotional exposure of citizens takes place during the electoral campaign, it is also relevant to analyze the role that leaders play in it and, therefore, to include them in our analysis. Thus, it is appropriate to explore the connection that leaders—i.e., candidates—try to generate with the electorate (Guo et al. 2012; Lynggaard 2019; Pereira et al. 2021); therefore, issues such as their facial expressions, the clothes they choose to wear at different events, or the people they surround themselves with are of great importance, as they try to generate emotional connections with citizens (Gravante and Poma 2018).

This whole complex landscape has been intensified by social networks. Thanks to them, the evolution of electoral campaigns has undergone significant changes and an increase in the use of materials with negative and/or satiric content has been observed (López-Paredes et al. 2022). This has been exacerbated by the entry of radical right-wing parties into campaigns (Gregor and Mlejnková 2021) and their recurrent recourse to what some authors have called 'LOLitics'—i.e., humorous catchphrases, videos or memes (Klein 2019). Again, emotions play a dominant role in these elements and, as a result, the use of negative campaigning seeks to appeal to factors related to fear by portraying the opponent as a threat and, in turn, presenting the promoter of the negative campaign as the solution to the problems (Witte and Allen 2000). The same is true when ridicule and humor are used against the opponent, although the latter approach can also be used to bring a distant candidate closer to the people (Voss 2009).

Social networks play a key role in electoral campaigns both because of their social penetration and their particular 'echo chamber effect' (Larsson 2020; Jaráiz Gulías et al. 2021; Rivera-Otero et al. 2021a; Agüero-Merino 2023). This phenomenon, which has been extensively studied in the academic literature, alludes to its similarity to the acoustic phenomenon in which a sound is repeated when sound waves hit an obstacle. Citizens commonly employ social networks to search for political information and participate in online political debates. These platforms feed on real-world events and are shaped by users to disseminate their own streams of thought and receive like-minded feedback. Similarly, in social networks, individuals usually interact with other users who share similar thoughts or behaviors. This happens because social networks are designed to be accessible to everyone, without excessive complications, to register and share states, feelings, tasks, and other thoughts.

It is important to note, according to Rivera-Otero et al. (2021a), that not all social networks offer the same services or are used in the same way, since each is specialized according to the type of content shared in it. However, one of the main reasons for the popularity and spread of social networks is the freedom they offer to publish a wide variety of content, albeit subject to certain legal restrictions, in different formats, ranging from videos and images to texts and voice messages. In addition, these platforms introduce additional elements to make messages more attractive, such as emoticons, stickers, GIFs and even hashtags. All of this makes it easier to spread the message and determines the way it is communicated.

This content ultimately generates a series of reactions in individuals; it is therefore necessary to analyze the impact of emotions and how they influence users' perceptions of the topics discussed in social networks. In addition to the echo chamber effect, it is also important to consider what Pariser (2012) calls 'filter bubbles'. These are generated because of the algorithms that operate in social networks, i.e., formulas that classify content to offer each user a personalized experience based on their usual consumption and previous searches. Unlike echo chambers, which only guarantee feedback with similar content, filter bubbles buttress people's attitudes by avoiding the consumption of information that contradicts their ideas, thus reinforcing their own thoughts and reducing their tolerance of what is different. Emotions play a fundamental role in social networks and shape the socialization of individuals, especially in the field of political information where they provoke intense reactions that generate heated debates, discord and an even greater strengthening of everyone's convictions.

In terms of credibility, social networks are the least credible medium compared to other more traditional media such as radio, newspapers or television (Rivera-Otero et al. 2021a; Gill 2022). This is mainly due to the difficulty of verifying the origin of the news read on social networks; it is sometimes difficult to know whether the information conveyed by an account is modulated by an external political influence.

These elements clearly influence the degree of affective polarization in society; in this context, there is evidence to suggest that Spain is one of the most polarized (Gidron et al. 2020). This situation is particularly pronounced in social networks and the digital sphere and does not fluctuate in the same way according to the platform used. Thanks to an analysis of the

'old Twitter', we can see how filter bubbles work (Yarchi et al. 2021). On Twitter, all content was freely available, although the personalized newsfeed was shaped each user's choice of who to follow. At the same time, users can also search for content using hashtags, which can cut across different political viewpoints (Kligler-Vilenchik et al. 2020). Despite this, we should not reject the role that social media can play in depolarizing society, especially in terms of affective polarization (Kubin and von Sikorski 2023). However, this is beyond the scope of this research as we do not want to cover elements such as the profiles that access the content analyzed here, or the level and direction of existing engagement, which, despite the relevance of these studies, cannot be analyzed with the data collected, as has been shown.

## 3. Electoral Context in Castilla y León and Andalusia

As mentioned, our focus is on the two elections that took place in 2022 (Gidron et al. 2020) in Castilla y León and Andalusia. Both, as we shall see, had their own peculiarities but they had in common the fact that they opened a new electoral cycle in Spain.

The study of these two regions is uneven in the academic literature. On the one hand, Andalusia is one of the most extensively analyzed regions from a political-electoral perspective (Ortega 2003; Montabes 2010; Trujillo and Ortega 2015). On the other, Castilla y León is one of the least analyzed as neither its party system nor its influence on national politics has been significant; studies have focused on its structure only in a general way (Fernández-Esquer and Dueñas-Castrillo 2018; Antón-Merino and Pérez-Castaños 2019).

An assessment of the electoral campaigns in Andalusia and Castilla y León has thus become a fundamental touchstone for understanding the evolution of those parties at the national level which operate at the extremes of the ideological spectrum. In the case of VOX, it was in the 2018 Andalusian elections that they entered a legislative chamber for the first time; already in 2022, they managed to improve their results by two representatives. For the first time in Spain, this populist radical-right party came to power in the regional government of Castilla y León, becoming the third electoral force in 2022 and forming a coalition government with the conservative Popular Party.

At the other end of the ideological spectrum, we find Podemos, or the UP coalition, a party that emerged in 2014 and, despite reaching historic highs in 2019, has seen its electoral weight decline. In the 2022 elections, it only managed to secure one representative in Castilla y León. In Andalusia, its brand obtained 17 representatives in 2018; by 2022, the coalition was divided into two parties: AA with two representatives and PORA with five[2].

It is therefore important to identify how these two organisations—that, since their creation, have sought to give voice to a section of the electorate dissatisfied with the existing party system and seeking a radical change in the way politics is conducted—are focusing their campaigns. They are also the key to governance in a fragmented party system.

In sum, it is worth noting that, in the case of the candidacies studied, both parties in Castilla y León presented men as candidates for the regional presidency. In the case of VOX, Juan García-Gallardo, who was running for the regional government for the first time; in the case of UP, Pablo Fernández, who had already been a candidate in 2015 and 2019. In the case of Andalusia, the three candidacies of the parties under analysis were led by women. In the case of VOX, Macarena Olona, leader of the party and national deputy. In the case of the radical left parties, PORA presented Inmaculada Nieto, Podemos' spokesperson in the Andalusian Parliament since 2019; AA, for its part, presented Teresa Rodríguez, member of the European Parliament and Podemos candidate for the regional presidency in 2018.

## 4. Methodology

This study has a double perspective. On the one hand, we show the electoral evolution of the extreme parties in Castilla y León and Andalusia and conduct a review of the X campaign of these five organisations, specified in both territories. This analysis has been carried out following a coding system, adapted from the EEMC[3] for the European Parliament elections in 2019, which was previously refined and tested by Pérez-Castaños et al. (2023) in the 2022 Castilla y León elections.

Table 1 shows that the populist radical-right party VOX does not change its name depending on the territory in which they seek office. However, in the case of the left-wing group Podemos, which is currently in the national government in coalition with the PSOE (Spanish Socialist Workers' Party), the name changes between scenarios. Thus, in the case of Castilla y León, the party is called the same as its national matrix; in Andalusia, as said, there is a split. In the 2018 elections, the AA electoral coalition—which included Podemos and its anti-capitalist split—presented itself as the only organisation to the left of the PSOE. In 2022, on the other hand, there is a split[4]. The anti-capitalist group, which broke away from Podemos, will keep the coalition's acronym while Podemos and other left-wing parties will be known separately as PORA[5].

**Table 1.** Political parties and denominations analyzed in both regions.

| Political Party (Ideology) | Denomination in Andalusia | Denomination in Castilla y León |
|---|---|---|
| VOX (Radical Right) | VOX | VOX |
| Podemos (Radical Left) | PORA | UP |
| Former-Podemos (Anticapitalist) | AA | - |

With these divisions in mind, the analyzed X accounts are presented in Table 2. They represent both the official accounts of the parties or coalitions as well as the people who exercised the candidacy for the regional presidency of these organisations. This adds up to 3097 tweets included in the analysis, of which 809 are from Castilla y León and 2288 from Andalusia.

**Table 2.** Candidates, X users and number of tweets during the election campaign.

| Region | Political Party | #Tweets | Candidate | #Tweets |
|---|---|---|---|---|
| Castilla y León | VOX | 64 | Juan García Gallardo | 29 |
| | UP | 197 | Pablo Fernández | 70 |
| Andalusia | VOX | 504 | Macarena Olona | 188 |
| | PORA | 1057 | Inmaculada Nieto | 181 |
| | AA | 588 | Teresa Rodríguez | 341 |

For its measurement, the codebook of Novelli and Johansson (2019), which refers to the European electoral campaign and of which we have already spoken, has been adapted. In addition, its codification has been adapted to a different territorial situation, including elements and themes that better fit the electoral arena analyzed. Likewise, the elements of measurement have been refined to adapt them to the Spanish reality.

As can be seen, the first major difference we found in our data is that the X campaign in Castilla y León was much less extensive than in Andalusia. This could be due, among other reasons, to the fact that the population of the Castilian-Leonese region is much more dispersed and older than that of Andalusia (Bello-Paredes 2020); although it could also be a decision by the territorial leadership of the parties, based on the contextual reality, or by the candidates themselves. As a result, the penetration of social networks has been lower and the use of traditional campaign formats such as television advertising, radio or face-to-face formats has been more relevant[6].

Before presenting the results, the coding of the items must be validated. In this case, it was carried out by the authors of this research themselves. To check the validity and reliability of the coding, three months after the initial coding, 10% of the materials were selected for re-encoding. The kappa index proposed by Cohen (1960) was used to carry out this coding. The equation for *K* is:

$$K = \frac{P_0 - P_e}{1 - P_e}$$

where $P_0$ is the relative observed agreement between observers and $P_e$ is the hypothetical probability of agreement by chance, using the observed data to calculate the probabilities of each observer randomly classifying each category. If the rates are in perfect agreement, then $K = 1$. If there is no agreement between the rates, other than what would be expected by chance, then $K = 0$. In this case, a result of 0.87 was obtained so it can be said that there is no bias in the coding process.

Thus, the aim of this article is none other than to investigate whether the emotional strategies of the far-right and far-left parties in the 2022 regional elections are different, paying special attention to the communication strategy of leaders and parties on X during the election campaign. As said, we expect to find differences between political parties but also between regions.

Thanks to the extracted data and its subsequent coding, a double investigation can be conducted. The first focuses on identifying, in a comparative way, how the campaign was distributed in the elements mentioned in the previous sections in both Autonomous Communities. The second, related to the emotional and rational aspects of the campaign in the social networks, will verify the differences not only among regions but also, within each one, between the political parties. This research, therefore, aims to test the following hypotheses:

**H1.** *VOX will use more emotional than rational content in both scenarios.*

**H1.1.** *Within this emotional content, negative emotions will be more predominant.*

To the extent that Spain is one of the most emotionally polarized societies (Gidron et al. 2020), it seems clear that the use of emotional content will be the most common in electoral campaigns that also open a new electoral cycle in the country.

**H2.** *Left-wing parties will make more use of emotional content.*

**H2.1.** *The emotions used will generally be more positive.*

Several authors have found that emotional content is more present in radical left parties than in radical right parties (Pliskin et al. 2014). In this sense, we hope that in this research we will find the same results as those already found in other scenarios.

**H3.** *VOX candidacies in both territories will use more negative and humorous content than left-wing parties.*

Although populist parties of the radical-right have changed their dialectics to broaden the electoral spectrum of possible voters to reach a larger audience (Borbáth and Gessler 2023, studies such as that of Novelli and Johansson 2019) detect a greater use of negative content in organisations of this type.

**H4.** *Leadership representation will be more formal in right-wing parties than in left-wing parties.*

**H4.1.** *Left-wing parties will tend to represent their leaders more closely.*

In this sense, some authors have noted that the image that populist far-right parties try to convey on social networks has for some time been linked to the projection of professionalism (Bast 2021); in addition, other authors (Duerksen and Elias 2018) have noted that populist left-wing leaders tend to present themselves as closer, more expressive and more emotional than the rest.

All these hypotheses will be tested by means of a descriptive study of the content of the materials posted on the social network X during the campaign period under analysis. In this way, the tests will be carried out in an exploratory manner as the data obtained do not allow for an in-depth statistical analysis due to the lack of statistical variables for this purpose.

## 5. Results

The first analysis refers to the percentage of content distribution in relation to the total number of tweets. Figure 1 shows the weight of each of the analyzed political parties in

each of the Autonomous Communities. Thus, the first observation we can make is how, within each region, left-wing political parties have a greater weight than that of VOX.

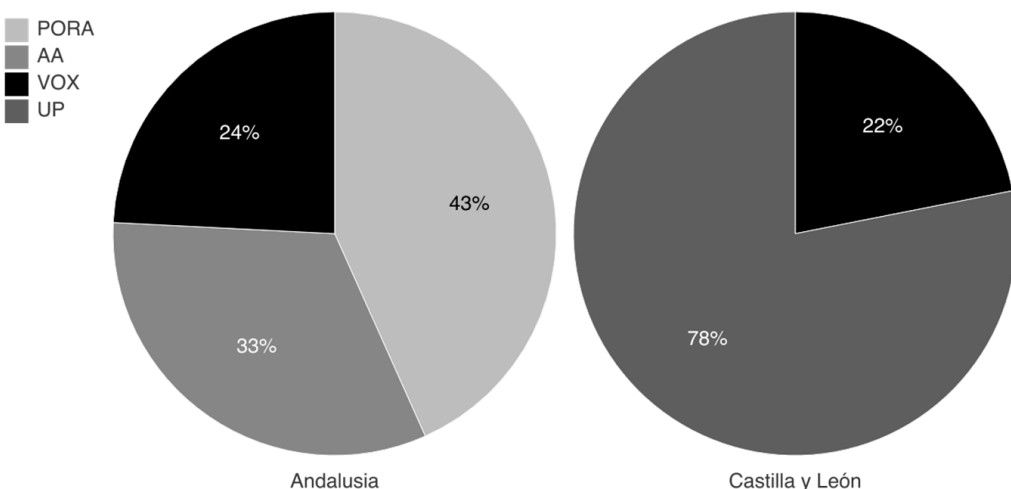

**Figure 1.** Tweets from each party per region (%).

In both cases, the radical-right parties report between 22 and 24% of the tweets; for the radical-left they are between 76 and 78%. In Andalusia, this percentage is divided between the two organisations, with PORA having the greater weight accumulating 43% of the total tweets.

Regarding the type of content shared and posted on social networks, Table 3 helps identify whether the content generated by these parties is their own or, on the contrary, they prefer to use content generated by other users and share it on their own profiles. It should be clarified that the shared content considers those tweets in which the accounts do not include new elements to the original.

**Table 3.** Shared and original content by party per region (%).

| Party | Andalusia | | Castilla y León | |
|---|---|---|---|---|
| | Shared | Original | Shared | Original |
| VOX | 74.1% | 25.9% | 27.7% | 72.3% |
| PORA | 54.4% | 45.6% | | |
| AA | 52.7% | 47.3% | | |
| UP | | | 4.2% | 95.8% |

The data show great differences among regions but also between the parties themselves. Starting with the first, we can see that in Castilla y León, original content is the most common, accounting for almost three out of four tweets from VOX and almost all of UP's content. In Andalusia, however, it is the shared content that predominates. In the case of VOX, the figures are the opposite, where three-quarters of the content is shared. In the case of the two left-wing parties, we can also see that, in both cases, the shared content predominates, although the figures are very similar, with AA having closer levels between the two elements.

Figure 2, on the other hand, shows us data on the type of material posted on X. This time, for the sake of clarity, all hits have been grouped together, with the first three corresponding to Andalusia and the next two to Castilla y León.

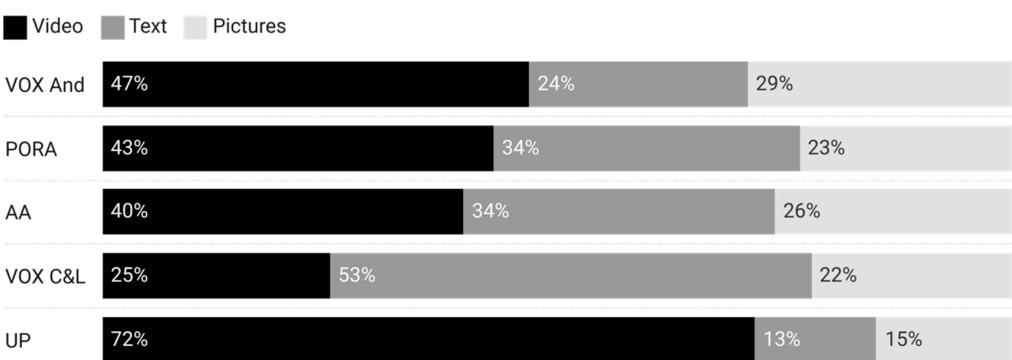

**Figure 2.** Type of material posted (%).

The data show how there are differences, again by region and between parties. In the case of Andalusia, all the parties show a clear trend towards video materials, with content between 40 and 47% of this typology. As for the following options, differences can be seen, with text materials being the favorite of both leftist parties (34%) compared to image materials, which is the favorite of VOX in the region (29%). In the case of Castilla y León, three out of four UP posts have a video, while photos and text are almost equally distributed; however, VOX, prefers to use text extensively (53%) while the distribution of audio-visual content is almost the same.

We now turn to the presence of leaders. To do this, Figure 3 shows the data, again segmented by region and political party, to achieve a greater width of analysis. The data show that it is very common to have leaders present in campaign materials. It should be clarified that the political leader was always the candidate for the presidency of the region or the national leader of the party.

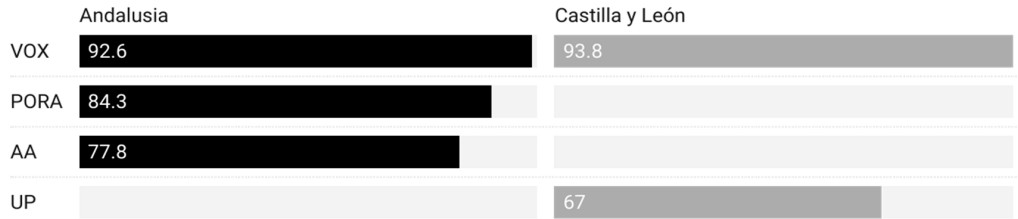

**Figure 3.** Materials showing leaders (%).

In this case, the differences are insignificant. In both territories, VOX has clear leadership in the content of X, although the leftist parties in Andalusia do not lag far behind with 84.3% and 77.8% of materials containing the leader in the cases of PORA and AA, respectively. In the case of UP in Castilla y León, the percentages are lower, limited to two out of three tweets with the presence of a leader.

The presence of leaders is more pronounced in populist radical-right parties, but what kind of leaders are present? Table 4 shows a disaggregation according to whether the leaders present are regional, national, a combination of both or even foreign. As before, the data for the three Andalusian organisations are presented first, followed by the data for the parties in Castilla y León.

In the case of AA, we observe that regional leaders, that is, of the Regional Government of Andalusia (Junta de Andalucía), or the candidates for the different constituencies are almost exclusively present. The leader appears in this category in 94.2% of the materials. The national leaders have little weight (2.2%) because this party had separated from Podemos. In PORA's data, it is appreciated how the presence of national leaders is the highest at 6.6% and almost a quarter of their materials (23.7%) have a combined presence of regional and national leaders. It is noteworthy that VOX in Andalusia is the only party to have an international presence, namely Georgia Meloni, Prime Minister of Italy. What is

more, her role is already common in all the electoral processes that have taken place since then, whether online or in person.

**Table 4.** Type of leader present (%).

| Party | Regional | National | Regional & National | Foreign |
|---|---|---|---|---|
| VOX And | 72.6% | 2.7% | 23.0% | 1.8% |
| PORA | 69.6% | 6.6% | 23.7% | 0.0% |
| AA | 94.2% | 2.2% | 3.6% | 0.0% |
| VOX C&L | 60.0% | 6.7% | 33.3% | 0.0% |
| UP | 91.4% | 4.3% | 4.3% | 0.0% |

If we look at Castilla y León, we can see that UP has greater regional personalism, with more than 90% of the materials representing the regional leader and obtaining a tie in those cases where a national leader and a combination of both leaders appeared (4.3% for each). VOX has the greatest joint presence of regional and national leaders. This can be explained by the poor knowledge level of the VOX candidate, García-Gallardo, and the need to support him with national leaders (33.3% of the content).

In addition to the presence of the leader, it is relevant how they are presented to the population through the clothes they wear and their facial expressions, as a way of transmitting emotions to citizens and presenting themselves as people who can be trusted. All this information could be found on Table 5.

**Table 5.** Dress code and leaders' expressions (%).

| Party | Dress Code | | | Expression | | | |
|---|---|---|---|---|---|---|---|
| | Formal | Formal | Casual | Smiling | Serious | Neutral | Funny |
| VOX And | 8.0% | 72.0% | 20.0% | 40.5% | 54.8% | 4.8% | 0.0% |
| PORA | 24.5% | 39.5% | 36.1% | 41.2% | 37.2% | 19.1% | 2.5% |
| AA | 0.9% | 27.8% | 71.3% | 40.4% | 38.6% | 21.1% | 0.0% |
| VOX C&L | 10.0% | 10.0% | 80.0% | 43.3% | 10.0% | 46.7% | 0.0% |
| UP | 11.3% | 68.3% | 20.4% | 26.3% | 66.1% | 5.9% | 1.6% |

This shows that there is no predetermined pattern between the political parties or differentiation according to the Autonomous Communities. In the case of VOX in Andalusia, we see that semi-formal dress is most common (in almost three out of four appearances); while in Castilla y León, however, its candidate is more casually dressed. As for the left-wing parties, in the case of Andalusia, PORA combines semi-formal and casual representation of its leaders, while AA shows Teresa Rodríguez in a more casual way. In the case of Castilla y León, UP mostly shows Pablo Fernández in a semi-formal way. These differences in VOX could be due to the different profile of the electorate in the area, the personality of the candidate, the socio-political moment and/or weather during the campaign[7] or the gender of the candidate. Dress codes do not show solid explanatory dynamics by party or region and can be linked to issues related to the image and personality of the specific leader.

In addition to the impact that clothing has on the population, facial expression is also of interest. In this sense, we see that closeness shown by a smile is one of the most common contents, except for UP in Castilla y León. In the case of VOX, however, seriousness is the most common expression in Andalusia (54.8%) and neutrality in Castilla y León (46.7%). Seriousness is also the expression that UP seems to present the most, which differentiates it from its Andalusian counterparts.

Figure 4 shows the start of the analyses of emotionality, rationality and the presence of negative and humorous content in social networks during the campaign. In this case, the first moment explores the presence of emotional, rational or neutral content by a political party. Figure 4 presents five graphs, the first three of which correspond to the Andalusian political parties and the last two to Castilla y León.

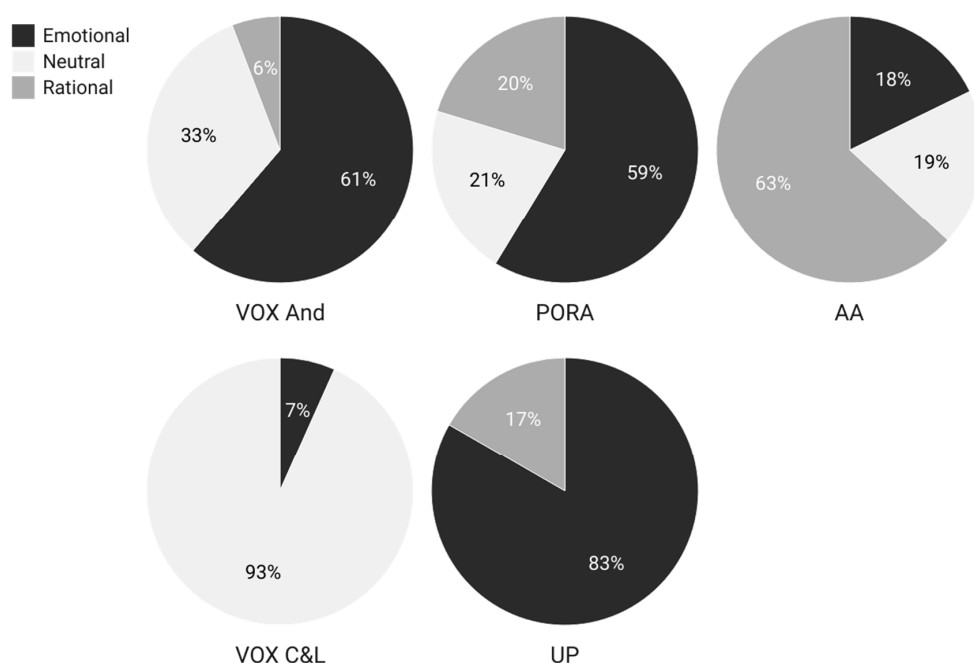

**Figure 4.** Type of content of tweets (%).

Emotional content is the most common for VOX (62%) and PORA (59%) in Andalusia, and for UP (83%) in Castilla y León. In the case of AA, rational content predominates (63%). Only in the case of VOX in Castilla y León is the majority of content neutral, representing almost all the tweets analyzed (93%)[8].

To deepen this, Table 6 refers to the type of emotion that these organisations represent in their material. A total of eight different emotions were measured: entertainment, anger, compassion, empathy/sympathy, enthusiasm, fear, pride and well-being. Of these, anger and fear are classified as negative emotions, while the others as positive. It should be noted that, in the cases under analysis, no messages related to fear were found so they do not appear in Table 6.

**Table 6.** Type of emotion by party (%).

| Party | Entert. | Anger | Comp. | Symp. | Enth. | Pride | Well. | Neg. | Pos. |
|-------|---------|-------|-------|-------|-------|-------|-------|------|------|
| VOX | 2.0% | 3.9% | 2.0% | 9.8% | 35.3% | 7.8% | 39.2% | 3.9% | 96.1% |
| PORA | 2.4% | 7.3% | 1.6% | 20.2% | 32.3% | 26.6% | 9.7% | 7.3% | 92.7% |
| AA | 0.0% | 6.1% | 3.0% | 15.2% | 30.3% | 27.3% | 18.2% | 6.1% | 93.9% |
| VOX | 0.0% | 0.0% | 0.0% | 100.0% | 0.0% | 0.0% | 0.0% | 0.0% | 100.0% |
| UP | 21.8% | 10.9% | 0.0% | 32.7% | 29.1% | 0.0% | 5.5% | 10.9% | 89.1% |

We can see that in the case of VOX in Andalusia, the most common emotion is well-being or feeling good (39.2%), followed by enthusiasm (35.3%). This is also the most common emotion for both PORA (32.3%) and AA (30.3%). In both cases, the second most common is pride. In the case of Castilla y León, we can see how the emotions change. On the one hand, we find VOX, where, as already mentioned, emotional content is very rare and, in this case, every element concentrates on the same emotion of sympathy. On the other, the left shows a greater variety of emotions, with data very similar to those we highlighted in Andalusia, in which sympathy is the emotion most employed (32.7%) followed by enthusiasm (19.1%) and entertainment (21.8%). In terms of positive and negative emotions, the former is the most predominant, comprising a total of 90% of materials. As for the negative emotions, it can be seen how UP in Castilla y León has most used this type of emotional content.

Finally, Figure 5 focuses on the use of satirical and negative content[9] in the campaign materials posted on X by the parties under analysis. These elements are not related to the emotions that they try to express in their tweets and that we have depicted in Table 5, but are oriented towards the object of the campaign; that is, they try to measure whether someone has been attacked or whether something or someone has been mocked or affected by satire. Again, to facilitate interpretation, the hits are presented consecutively: first the three Andalusian hits and then the two Castilian-Leonese.

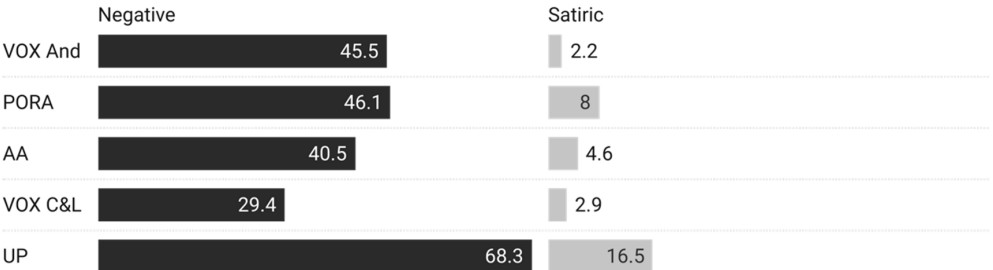

**Figure 5.** Negative and satirical content by party (%).

As can be seen, there are significant differences by region and, in some cases, by ideology. The first thing to note is the scarcity of satirical content, with less than 10% of materials trying to make fun of the media, rival leaders or people in general. A different case is that of UP in Castilla y León, where we find 16.5% of materials oriented towards this type of campaign. They are followed by PORA in Andalusia (8%) and then AA (4.6%). VOX, in both Andalusia and Castilla y León, has barely 3% of this type of content. This seems to indicate that the extreme-left is more likely to use satirical or humorous material in its campaign than the extreme-right.

Turning to negative content, there is no clear trend in any of the perspectives analyzed in this study. If we look at the parties in Castilla y León, we can see that UP used negative campaigning more than VOX, to the extent that 68.3% of the materials on X had this orientation for the extreme-left, while only 29.4% of VOX's content fell into this category. In the case of Andalusia, VOX and PORA have similar percentages (45.5% and 46.1%, respectively), followed by AA with 40.5% of the content. These data seem to reflect that the Andalusian campaign had a more negative perspective on the part of the extreme parties.

## 6. Discussion

The results now allow us to discuss the content of the social networks and, therefore, to test the hypotheses put forward in the methodological section of this work. However, before going into the details, it is worth highlighting the role that the different parties played in the campaign on social networks. Thus, in Andalusia, tweets from the left have accounted for more than 75% of the content analyzed, leaving a quarter of the content to the populist radical right, which advances the postulates of Pliskin et al. (2014) on the more extensive and active presence of far-left parties in social networks. This trend is highlighted for the Andalusian case, where radical-left parties are those that generate more original content on social networks, while populist radical-right parties have a higher percentage of shared content. In the case of Castilla y León, the first of these trends is identical, perhaps a result, among other things, of the demographic situation in which the region finds itself, as has previously been pointed out (Bello-Paredes 2020). However, contrary to what happened in Andalusia, here we find more original content in the VOX account and more shared content in the UP account, which acts in a counterintuitive way to what is indicated by the theory.

Starting with the first hypothesis, we expected to find more emotional content in VOX's social media material and that this emotional content would also have a more negative aspect. In this sense, we see that VOX's content is more emotional in the case of Andalusia, but not so in Castilla y León. However, in both cases, the choice is to use videos

on social networks, which help to disseminate the content in a more emotional way. In terms of the emotional typology we find, we can confirm that the emotions they convey are not negative, but quite the opposite. This tendency is found in all the parties since most of the emotions that the left-wing and radical right-wing parties try to transmit on social networks have a broadly positive tone. These results are surprising in the sense that, as mentioned by Gidron et al. (2020), Spain is one of the most emotionally polarized Western countries.

Secondly, we tried to verify the content of the campaign of the radical-left to find out if its emotional content is more extensive than the rational one; in addition, we expected to find positive emotions to a greater extent, in line with those established by Duerksen and Elias (2018). In this sense, as mentioned, the positive emotional content is in fact the most widely used, regardless of the extremity of the ideological spectrum in which they are located. So, we consider the existing sub-hypothesis to be proven. Regarding the first hypothesis, it is true that the emotional content is more present in the orgnisations that have integrated Podemos, such as UP in Castilla y León and PORA in Andalusia; this condition is not met for the anti-capitalist party AA. Furthermore, in Castilla y León, they shared more audio-visual content in networks but not in Andalusia, where the textual content was more extensive.

The third objective we set ourselves was to prove that the content of VOX would contain more attacks, either negative or humorous. Despite the findings of Borbáth and Gessler (2023), who note that the rhetoric of populist far-right parties has changed as they seek to expand their electoral base, in this case we find that this hypothesis is proven, since it is in Andalusia where negative content is most used by VOX, in line with the findings of Novelli and Johansson (2019). However, regarding the satirical or humorous attack, we see that this issue is typical of far-left parties, in accordance with what has already been commented upon and analyzed by Duerksen and Elias (2018). In this way, we could consider the third part of the hypothesis partially true, highlighting the fact that the first half is only fulfilled in Andalusia and not in Castilla y León.

To conclude the testing of the hypotheses, the fourth sought to examine how the populist radical-right parties present their leaders, expecting to see them more formally dressed. Based on this, we proposed a sub-hypothesis focusing on the fact that the radical-left parties would try to present their leaders in a more intimate way. The data we collected show that there is a greater presence of leaders in the content posted on X by the populist radical-right. In the case of VOX, we also find differences based on the Autonomous Community, to the extent that they represent the most formal and serious leaders in Andalusia—in line with what was revealed by Bast (2021)—while the representation of these in Castilla y León is closer and more casual. For its part, the representation of the leaders of the extreme-left in Andalusia conforms to the criteria of Duerksen and Elias (2018), with them being more casual and closer; it does not do so in Castilla y León, where the UP candidate appeared dressed in a more formal way and with a more serious facial expression. However, it can be said that the fourth hypothesis is only fully met in the case of the extremist organisations in Andalusia.

Summarizing the fulfilment or non-fulfilment of the hypotheses, we can say that H1 is partially correct in the case of Andalusia but not in the case of Castilla y León. On the other hand, H1.1 is not correct, with positive emotions being the most predominant. As for the second set of hypotheses, H2 is true in the case of the Podemos parties—UP in Castilla y León and PORA in Andalusia—but not in the case of the anti-capitalist split of Podemos in the latter region. H2.1 proved correct in all cases. H3, which refers to the negative and humorous attack, turned out to be false, with the sole exception of the negative content for VOX in Andalusia. Finally, in the fourth block, H4 has been shown to be partially correct—for VOX in Andalusia, but not in Castilla y León—while H4.1 has followed the same dynamic, i.e., it is correct for Andalusian parties but not for those in Castilla y León. The implications of these findings are detailed in the conclusions.

On the other hand, it should be noted that there are important differences between the two regions. In the case of Castilla y León, for example, only the hypotheses related to UP are tested in terms of the use of emotions. The remaining hypotheses go in the opposite direction to that indicated by the theoretical rules already discussed. In the case of Andalusia, the dynamics seem to be different, with the only differences found being the absence of negative emotions in VOX—an element already detected by Borbáth and Gessler (2023)—and the appeal to more rational content by the radical-left party, AA.

## 7. Conclusions

The results enable us to respond to the first objective of this research, which was to identify whether there are differentiated emotional regimes between the left-wing and populist radical-right parties. We can answer this question not only by saying that there are differences, but that they exist not only among the parties of the left and the radical right, but also between the Autonomous Communities in which they compete.

Thus, in the case of VOX, the predominance of emotional content occurred in Andalusia but not in Castilla y León, where the most common content in social networks was neutral or rational. Regarding this emotional content, it is worth highlighting the extensive use of positive emotions is perhaps in line with this search to expand its electoral base, as analyzed by Borbáth and Gessler (2023) in the case of Hungary. On the radical-left, the organisations in which Podemos was integrated also show a greater emotional content than the split (AA) with an anti-capitalist aspect. However, in all three parties, positive emotional content is again the most common. This shift marks a new trend that will have to be tested in future analyses, which may establish whether this is anomalous contextual behavior or whether it is due to a new moment in the electoral campaign of the radical-left and right parties.

The main differences among ideological groups can be found in the presence of attacks on other parties, people, institutions, etc. In the case of populist radical parties, the use of negative content against third parties was more common than in the case of left-wing parties. Even so, it should be made clear that, in general, negative content was used more by Andalusian organisations than by Castilian-Leonese ones. Moreover, in this latter community, the roles are again reversed, with UP making more use of a negative campaign. This may be due to UP's gradual withdrawal from the institutions in the region, which may have led to a "radicalization" of its campaign strategy; this may also explain the fact that UP's campaign in Castilla y León has developed contrary to what previous studies had assumed.

However, when it comes to satirical attacks, there are differences between the extreme-right and extreme-left. The former is less inclined to use this type of satirical attack, while the latter have used it more extensively. Nonetheless, it should be made clear that this type of action is unusual in the Spanish electoral campaign.

Undoubtedly, this study leaves us with several questions that will have to be answered later in different and deeper analyses. The first refers to the counter-tendency of the electoral campaign in Castilla y León, not fulfilling any of the precepts that the theory seemed to suggest. But weight to all, this region is the first place in Spain where VOX managed to enter positions of institutional representation, obtaining the presidency of the regional parliament and with a vice-presidency of the regional government as well as with three regional ministries. It is left to future research to identify the reasons for VOX's success and if, as everything seems to indicate, it has benefitted from acting counterintuitively.

Another element that needs to be explored in more depth is the gender of the communicator to check whether it produces differences in the way citizens perceive the information they find on social networks. In this sense, Ruiloba-Núñez and de Zuazu (2022) consider this element to be relevant, showing how issues such as stereotypes, which still exist in terms of the characteristics commonly associated with each gender, are still influential. This assumption about the characteristics of female leadership and its differences from those of males, although it has been looked at by some studies and the term 'feminization' of politics

has been established, is based on the conclusion of the authors that there are no aspects of women that define them as a differentiated group in communication and legislation. This is not incompatible with accepting that women may be more inclined to certain types of action, but not exclusively, so that they should be treated as specific actors and not included as a gendered group (Ruiloba-Núñez and Samaniego 2021). This is particularly interesting given the fact that all the candidates in Castilla y León were men while in Andalusia all the candidates were women.

A third line of study that should be exploited is the use of additional elements to make messages more attractive, such as emoticons, stickers, GIFs and even hashtags, all of which are common in the case of X. These types of elements facilitate the dissemination of the desired message and determine the way in which it is communicated, in addition to conveying emotions that go beyond and can even be contrary to those conveyed by other contents of the same messages on X.

However, there are great differences between the extremes of the ideological spectrum, even though appealing to emotional elements was a leitmotif in all scenarios. Thus, even though electoral campaigns are elements that have been widely studied in the Social Sciences in general and in Political Science in particular, it seems that an analysis of what is on offer in campaign materials and social networks will lead us to repeat the title of the 1960 film *Once more, with feeling!*.

**Author Contributions:** Each author contributed equally to all aspects of the article. All authors have read and agreed to the published version of the manuscript.

**Funding:** This research received no external funding.

**Informed Consent Statement:** Not applicable.

**Data Availability Statement:** The graphics used in this article are the data and are fully available for anyone to access. If further explanations on the data are needed, please contact the corresponding author.

**Conflicts of Interest:** The authors declare no conflict of interest.

## Notes

[1] Many authors have described VOX as a populist radical right party that has increased its electoral support since 2018 (Anduiza 2018; Antón-Merino et al. 2023).

[2] These parties' literal translation is for AA 'Forward Andalusia', and for PORA 'For Andalusia'.

[3] The EEMC, which stands for the European Elections Monitoring Centre, carried out a massive monitoring of the 2019 European elections in all the countries where they took place. Further information can be found on their website: https://www.electionsmonitoringcenter.eu (accessed on 24 June 2023).

[4] The result of an internal division that has already been reflected in the regional parliamentary seat since 2021.

[5] It was formed by Izquierda Unida Los Verdes-Por Andalucía, Podemos and Iniciativa del Pueblo Andaluz, Mas País Andalucía, Equo and Alianza Verde. All of them are political parties that belong to the radical left, environmentalist and communist ideologies, among others. More information about this coalition can be found on their website: http://www.porandalucia.org (accessed on 3 September 2022).

[6] However, these questions are not part of the study and would require a different type of approach. Therefore, they are set aside for the time being.

[7] This is noteworthy as the elections in Castilla y León took place in February, with a mean temperature during the campaign moving from 1 to 10 Celsius degrees; but in Andalusia, it took place during June, with a temperature moving from 18 to 33 Celsius degrees.

[8] It may be of interest in further work to identify differences in the comparative expression of emotions between these parties at the extremes and political groups at other points on the political spectrum (contrasting claims about the generality of polarization and actors' use of emotions across the political system or, conversely, extensive use between parties on the extreme right and left).

[9] By negative content, we mean messages that criticize the management of the previous government, use derogatory expressions or negative evaluations of the candidate of another party, develop arguments linked to emotions of weariness, etc.

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
