# Peer review of "Once More, with Feeling! Digital Campaigns and Emotional Candidacies in X in Andalusia and Castilla y León"

_socsci, doi:10.3390/socsci12090504_

Round 1

Reviewer 1 Report

The paper (Once More, with Feeling! Digital Campaigns and Emotional 2 Candidacies in Twitter in Andalusia and Castilla-y-León) is of interest to the special issue for two reasons. In the first place, because of the subject matter; secondly, by the methodology. Finally, for doing it in a comparative way between communities in Spain.

The status of the question is correct (introduction), but it is suggested to make changes. It is important to expand the "state of the problem", introducing more studies on emotions and social networks in the Spanish context. What were your results? What are your limitations? Likewise, it is important to differentiate the "state of the question" with the explanation or the political context in Andalusia and Castilla León. It is suggested to introduce a different context point. I also have doubts about placing Podemos and Vox at the same level of "populism". Further academic and scientific support for this statement is suggested.

In the theoretical part, it is also interesting to work more on the concept of "polarization" caused by emotions in social networks. For example, the analysis of political polarization is approached, theoretically and empirically, from three areas (Yarchi et al., 2021): polarization in interaction (between different digital communities); positional polarization (between the different positions expressed) and affective polarization (relative to emotions). The methodological section is correct. The hypotheses are correct and the method of analysis of emotions is correct. Perhaps it is appropriate to ask research questions and not hypotheses. The validation of the instrument is valid as well as the analysis codes (Novelli and Johansson, 2019). The results are correct, although in their first part somewhat weak and very descriptive. Is it possible to introduce more complex indicators, such as engagement? Is it possible to introduce another variable (issues, for example) that allows a more complex emotional analysis? The conclusions and discussion are correct.

Author Response

First of all, we would like to thank reviewer 1 for his comments, which will undoubtedly improve the manuscript. With regard to the various points raised, we have made the following changes:

  1. We have added two new sections to the manuscript. One on the literature review and another on the electoral context. This is to expand on the state of the problem in the introduction, but also to expand on polarisation in the literature review section.
  2. We treat the element concerning VOX and Podemos on the same level of populism. This was never our intention, but we have made it clear. We are analysing the two extremes of the ideological spectrum, although both make extensive use of populist rhetoric in their messages. Since the main point of analysis is not the populist rhetoric but the emotional content, the analysis shows that both extremes move along the same path.
  3. We have changed the name "Twitter" to "X", which is the new name of the company and the application.
  4. We have decided to keep the terminology of "hypotheses", since these are hypotheses that will be tested thanks to the analysis carried out. Nevertheless, we have strengthened how we are going to check these hypotheses, whether the hypothesis is tested or not and what are the implications of the results.
  5. Regarding the introduction of deeper statistical analysis or other elements in the study, we are sorry to inform you that at the moment we do not have this kind of information and since Twitter (now X) has not made the academic tools free, it is not possible for us to access any kind of data. Nevertheless, we are trying to deepen the coding for the next researches, as the recommendation to use different topics is very helpful.
  6. We have sent the document for an English revision, which delayed the review,

We would like to thank reviewer 1 for his comments and willingness to improve our manuscript.

Reviewer 2 Report

This is an interesting paper and will contribute to political communication scholarship. This paper used a social media content analysis technique to compare the emotional strategies of far-right and far-left parties in the 2022 regional elections, paying special attention to leaders’ and parties’ communication strategy on Twitter during the electoral campaign. The author(s) found more emotional content in VOX's social media material and that this emotional content but not as ‘dark’ as they had hypothesised in the Castile and Leon. The author(s) glossed over this difference in outcome in comparison to their stated hypothesis. The paper was well conceived. However, some minor points need to be addressed.

Introduction: The author(s) should check the last two sentences of the abstract. It is unclear to me what they are trying to say. In the introduction, the author(s) could sharpen their research problem a bit more. For instance, when did the introduction stop and where did the literature review begin? I would suggest that the author(s) distinguish between the “introduction” and the “literature review” sections. In its current form, the introduction is too long and is not punchy. The justification is too drawn out. The author(s) started making attempt at justifying the originality and contribution of their study on page 3 beginning in lines 137. That is not right. I would suggest that the author(s) revise the introduction and divide it into two for clarity and structural strength. On a minor note, the author(s) should also check the sentence starting from line 84.

Method and Results: Method is clear, but the author(s) did not expressly discuss how their hypotheses will be tested. I would like the author(s) to clearly delineate how their hypotheses will be tested. They should also justify their analytic technique. Currently, it appears that the author(s) used descriptive statistics to test their hypotheses. I would like to see a justification of such hypotheses testing.

Discussion: The author(s) should clearly state and discuss the implication of the result. For example, when discussing their hypotheses, the author(s) should categorically state if their hypotheses were supported or partially supported of was not supported by their findings. Their first hypothesis from their commentary was only partially supported, yet the author(s) did not critically discuss the implication. The author(s) noted: “starting with the first hypothesis, we hoped to find more emotional content in VOX's social media material and that this emotional content would also have a more negative aspect. In this sense, we see that VOX's content is more emotional in the case of Andalusia, but not so in Castile and Leon.” What does this mean? How does this impact your study’s findings and contribution. I say this because the author(s)

noted in the abstract thus, “We hope to find differences between political formations, but also between regions”. The author(s) should also discuss their hypotheses in a systematic fashion. For example, the author(s) jumped from hypothesis 1 to the 4th hypothesis. Where are hypotheses 2 and 3? The author(s) should check this.

This study will add to the literature on political communication and the role that emotions play in electioneering campaigns. However, the author(s) will need to tighten loose ends before it can be accepted.

There are no significant language problem in the study.

Author Response

First of all, we would like to thank the reviewer for his comments, which will undoubtedly improve the manuscript. With regard to the various points raised, we have made the following changes:

  1. We have added two new sections to the manuscript. One on the literature review and another on the electoral context. This is to expand on the state of the problem in the introduction, but also to move forward our justification, now on pages 1-2.
  2. We have reviewed the last two sentecens of our abstract.
  3. We have changed the name "Twitter" to "X", which is the new name of the company and the application.
  4. We have reviewed and rewritten the sentence starting on line 84.
  5. We have clearly stated how the hypothesis will be tested.
  6. We have clearly stated the implications of the results and also whether the hypothesis is tested or not, including comments on all four hypotheses and the differences between regions.
  7. We have sent the document for an English revision, which delayed the review.

We would like to thank the reviewer for his comments and willingness to improve our manuscript.